Diagnosis of bacterial vaginosis by a new multiplex peptide nucleic acid fluorescence in situ hybridization method

Machado António
Castro Joana
Cereija Tatiana
Almeida Carina
Cerca Nuno nunocerca@ceb.uminho.pt
Centre of Biological Engineering, LIBRO—Laboratory of Research in Biofilms Rosário Oliveira, University of Minho , Campus de Gualtar, Braga , Portugal
Smidt Hauke
Electronic publication date: 2015 Feb 17
Publication date: 2015
Volume: 3
Electronic Location ID: e780
Received 2014 Dec 15; Accepted 2015 Jan 30
Copyright: © 2015 Machado et al.
Copyright year: 2015
Copyright holder: Machado et al.
License: This is an open access article distributed under the terms of the Creative Commons Attribution License, which permits unrestricted use, distribution, reproduction and adaptation in any medium and for any purpose provided that it is properly attributed. For attribution, the original author(s), title, publication source (PeerJ) and either DOI or URL of the article must be cited.
License URL: https://creativecommons.org/licenses/by/4.0/

Keywords: Fluorescence in situ hybridization (FISH), Peptide Nucleic Acid Probe (PNA probe), Lactobacillus spp., Gardnerella vaginalis, Bacterial vaginosis, Biofilms

Funding: European Union funds (FEDER/COMPETE) National funds (FCT) PTDC/BIA-MIC/098228/2008 Programa Operacional Regional do Norte QREN FEDER Portuguese national funds This work was supported by European Union funds (FEDER/COMPETE) and by national funds (FCT) under the project FCOMP-01-0124-FEDER-008991 (PTDC/BIA-MIC/098228/2008). António Machado was funded by the FCT individual fellowship SFRH/BD/62375/2009. Other fundings came from the FCT Strategic Project PEst-OE/EQB/LA0023/2013, the Project NORTE-07-0124-FEDER-000027, co-funded by the Programa Operacional Regional do Norte (ON.2–O Novo Norte), QREN, FEDER, and the project FCOMP-01-0124-FEDER-008991 (RECI/BBB-EBI/0179/2012). Nuno Cerca is an Investigador FCT (Portuguese national funds). The funders had no role in study design, data collection and analysis, decision to publish, or preparation of the manuscript.

==============================
Bacterial vaginosis (BV) is one of most common vaginal infections. However, its diagnosis by classical methods reveals low specificity. Our goal was to evaluate the accuracy diagnosis of 150 vaginal samples with research gold standard methods and our Peptide Nucleic Acid (PNA) probes by Fluorescence in situ Hybridization (FISH) methodology. Also, we described the first PNA-FISH methodology for BV diagnosis, which provides results in approximately 3 h. The results showed a sensitivity of 84.6% (95% confidence interval (CI), from 64.3 to 95.0%) and a specificity of 97.6% (95% CI [92.6–99.4%]), demonstrating the higher specificity of the PNA-FISH method and showing false positive results in BV diagnosis commonly obtained by the classical methods. This methodology combines the specificity of PNA probes for Lactobacillus species and G. vaginalis visualization and the calculation of the microscopic field by Nugent score, allowing a trustful evaluation of the bacteria present in vaginal microflora and avoiding the occurrence of misleading diagnostics. Therefore, the PNA-FISH methodology represents a valuable alternative for BV diagnosis.

Introduction

Bacterial vaginosis (BV) often exhibits high prevalence, high relapse rates and associated complications, which renders this infection of global importance (Falagas, Betsi & Athanasiou, 2007; Tibaldi et al., 2009). BV is associated with increased taxonomic richness and diversity (Oakley et al., 2008) and is normally characterized by a decrease in vaginal lactobacilli and a simultaneous increase in the anaerobes population (Tibaldi et al., 2009). Therefore, vaginal bacterial communities differ dramatically between healthy patients and patients with BV, where G. vaginalis is present in over 90% of BV cases (Verstraelen & Swidsinski, 2013). The role of G. vaginalis is still controversial, since this bacterium is also present in 10–40% of healthy women (Aroutcheva et al., 2001; Hickey & Forney, 2014; Silva et al., 2014); however, recent evidence suggests that the presence of G. vaginalis biofilms, instead of dispersed cells, are in fact an indication of BV (Verstraelen & Swidsinski, 2013). Furthermore, we recently demonstrated that G. vaginalis has a significantly higher virulence potential than other 29 BV associated species, including a higher cytotoxicity and biofilm formation ability (Alves et al., 2014). We also provided evidence that G. vaginalis biofilms can establish synergistic relationships with other BV anaerobes (Machado, Jefferson & Cerca, 2013), further highlighting its pivotal role on BV development.

The most frequently used method for BV diagnosis is the physician’s assessment by the Amsel clinical criteria (Forsum, Hallén & Larsson, 2005). This method is fairly subjective and is based on the observation of the following symptoms: vaginal fluid with pH above 4.5; positive “whiff test” (detection of fishy odor upon 10% potassium hydrogen addition); presence of clue cells (vaginal epithelial cells covered by bacteria) on microscopic examination of vaginal fluid; and homogeneous milky vaginal discharge. At least three of the four symptoms described above must be present to establish a positive BV diagnosis (Amsel et al., 1983). Despite the fact that the Amsel criteria does not require intensive training, it is not the most appropriate method to diagnose BV, due to its low specificity (Dickey, Nailor & Sobel, 2009).

Alternatively, laboratory diagnosis is based on the Nugent score analysis, a microscopic method that quantifies three different bacteria morphotypes presented in the vaginal smears (Nugent, Krohn & Hillier, 1991). These authors have created a Gram stain scoring system based on the evaluation of the following morphotypes: large gram-positive rods (Lactobacillus spp. morphotypes); small gram-variable rods (G. vaginalis morphotypes); small gram-negative rods (Bacteroides spp. morphotypes); and curved gram-variable rods (Mobiluncus spp. morphotypes). Each morphotype is quantified from 0 to 4 with regard to the number of morphotypes observed in the microscopic fields of the Gram-stained vaginal smear. The vaginal microflora is then classified in normal microflora (scores of 0 to 3) or as BV (scores of 7 to 10), based on the sum of each morphotype score (Livengood, 2009; Nugent, Krohn & Hillier, 1991). However, the evaluation of smears is also subjective and user dependent (Sha et al., 2005). Furthermore, due to its low specificity, the Nugent method also considers intermediate microflora whenever the final score is between 4 and 6.

Although both methodologies are easy and fast to perform, they do not provide a robust diagnosis of BV. When combined, these standard tests have a sensitivity and specificity of 81 and 70% (Forsum, Hallén & Larsson, 2005), respectively. To improve BV diagnosis, several new molecular methodologies have been proposed, with fluorescence in situ hybridization (FISH) being a very promising alternative. This technique combines the simplicity of microscopic observation and the specificity of DNA/rRNA hybridization, allowing the detection of selected bacterial species and morphologic visualization (Justé, Thomma & Lievens, 2008; Nath, 2000). Peptide Nucleic Acid (PNA) probes are used instead of natural nucleic acids to improve FISH efficiency because they enable quicker and more specific hybridization (Lefmann et al., 2006; Peleg et al., 2009; Wilson et al., 2005). These types of probes, in which bases are linked by a neutral peptide backbone, avoid the repulsion between the negatively charged phosphate backbone characteristics of DNA/DNA hybridization (Stender et al., 2002). Since PNA is a synthetic molecule, probes are also resistant against cytoplasmic enzymes such as nucleases and proteases (Amann & Fuchs, 2008). In addition, the hybridization step can be performed efficiently under low salt concentrations, which promotes the destabilization of rRNA secondary structures and consequently improves the access to target sequences (Almeida et al., 2009; Cerqueira et al., 2008). All these advantages have made PNA-FISH a new promising tool for diagnosis and therapy-directing techniques, providing already a rapid and accurate diagnosis of several microbial infections (Hartmann et al., 2005; Shepard et al., 2008; Søgaard et al., 2007; Trnovsky et al., 2008).

We have previously developed a multiplex PNA-FISH method able to specifically quantify in vitro Lactobacilli spp. and G. vaginalis adhered to HeLa cells (Machado et al., 2013). To determine the feasibility of our novel PNA-FISH method as a diagnostic tool for BV, we have blind-tested our multiplex methodology on vaginal samples from Portuguese women and compared those results with the laboratory microscopic derived method using the Nugent score.

Material & Methods

Vaginal sample collection and preparation

A total of 200 vaginal fluid samples were obtained, after informed consent, as approved by the Institutional Review Board (Subcomissão de Ética para as Ciências da Vida e Saúde) of University of Minho (process SECVS 003/2013). The vaginal samples were collected for Gram staining and FISH procedures, using the culture swab transport system (VWR, CE0344, Italy). The extraction procedure from transport media was elaborated in accordance with Money’s guidelines to avoid misleading in the Nugent score analysis of the vaginal swabs (Money, 2005). These swabs were brushed against the lateral vaginal wall to collect the vaginal fluid sample, placed into the culture swab transport media and immediately stored at 4 °C. First, the vaginal samples were used for Gram stain procedure, as described by Nugent, Krohn & Hillier (1991). Next, swabs were immersed in 1 ml of phosphate buffer saline (PBS) and the remaining vaginal material collected by centrifugation at 17,000 g during 5 min at room temperature. Afterwards, the pellet was resuspended in 2 ml of saline solution (0.9%NaCl) and finally diluted 1:10 in saline solution or PBS to eliminate possible contaminants that could interfere with FISH procedures, as previously described (Machado et al., 2013).

Classification of vaginal smears according to Nugent score

Vaginal samples evaluation was performed using the Nugent criteria score (Nugent, Krohn & Hillier, 1991). Briefly, Gram stained vaginal smears were examined under oil immersion objective (1,000× magnification) and 10–15 microscopic fields were evaluated for each sample. The composite score was grouped into three categories, scores 0–3 being normal, 4–6 being intermediate, and 7–10 being definite bacterial vaginosis. Finally, the smears that showed scores between 0–3 and 7–10 were selected for further study, as normal (−) and BV (+) samples, respectively. Meanwhile, the smears with a Nugent score of 4–6 or with incomplete epidemiological data were rejected from our study.

Fluorescent in situ hybridization

The 150 BV+ or BV− (as described above) vaginal samples were used on a blind PNA-FISH test. For each sample, 20 µl of the final suspension were spread on glass slides. The slides were air-dried prior to fixation. Next, the smears were immersed in 4% (wt/vol) paraformaldehyde (Fisher Scientific, Leicestershire, United Kingdom) followed by 50% (vol/vol) ethanol (Fisher Scientific, Leicestershire, United Kingdom) for 10 min at room temperature on each solution. After the fixation step, the samples were covered with 20 µl of hybridization solution containing 10% (wt/vol) dextran sulphate (Fisher Scientific, United Kingdom), 10 mM NaCl (Sigma, Seelze, Germany), 30% (vol/vol) formamide (Fisher Scientific, Leicestershire, United Kingdom), 0.1% (wt/vol) sodium pyrophosphate (Fisher Scientific, Leicestershire, United Kingdom), 0.2% (wt/vol) polyvinylpyrrolidone (Sigma, Seelze, Germany), 0.2% (wt/vol) ficoll (Sigma, Seelze, Germany), 5 mM disodium EDTA (Sigma, Seelze, Germany), 0.1% (vol/vol) triton X-100 (Sigma, Seelze, Germany), 50 mM Tris-HCl (at pH 7.5; Sigma, Seelze, Germany) and 200 nM of each PNA probe (Lactobacillus spp. PNA Probe: Lac663 probe, Alexa Fluor 488-ACATGGAGTTCCACT; HPLC purified >90%; Gardnerella vaginalis PNA Probe: Gard162 probe, Alexa Fluor 594-CAGCATTACCACCCG; HPLC purified >90%). Subsequently, the smears were covered with coverslips and incubated in moist chambers at the hybridization temperature (60 °C) during 90 min. Next, the coverslips were removed and a washing step was performed by immersing the slides in a pre-warmed washing solution for 30 min at the same temperature of the hybridization step. This solution consisted of 5 mM Tris base (Fisher Scientific, Leicestershire, United Kingdom), 15 mM NaCl (Sigma, Seelze, Germany) and 0.1% (vol/vol) triton X-100 (at pH 10; Sigma, Seelze, Germany). Finally, the glass slides were allowed to air dry.

Fluorescence microscopic visualization and bacterial quantification

Prior to microscopy, one drop of non-fluorescent immersion oil (Merck, Darmstadt, Germany) was added to either slides and covered with coverslips. Microscopic visualization was performed using an Olympus BX51 (Olympus Optics Portugal SA, Lisboa, Portugal) epifluorescence microscope equipped with a CCD camera (DP72; Olympus, Shinjuku, Tokyo, Japan) and filters capable of detecting the two PNA probes (BP 470-490, FT500, LP 516 sensitive to the Alexa Fluor 488 molecule attached to the Lac663 probe and BP 530-550, FT 570, LP 591 sensitive to the Alexa Fluor 594 molecule attached to the Gard162 probe).

In each experimental assay, a negative control was performed simultaneously, in which all the steps described above were carried out, but where no probe was added in the hybridization step. Finally, 20 random regions of each glass slide were photographed. All images were acquired using Olympus CellB software using a total magnification of ×1,000.

Statistical analysis

The data was analyzed to calculate sensitivity, specificity, accuracy, positive and negative likelihood ratios (PLR and NLR, respectively) of the PNA-FISH methodology, with 95% confidence intervals (CI), using a clinical online statistical software (www.vassarstats.net/clin1.html; accessed 2014) (Senthilkumar, 2006). The classic Nugent criteria score was used as the diagnostic true.

Results and Discussion

On this prospective study, 150 vaginal samples were used to compare BV diagnosis by the classic Nugent criteria and our PNA-FISH methodology. As shown in Table 1, the main characteristics of the sample population used to validate our method mirrors what has been described in other main epidemiological studies, namely (1) the overall rate of positive BV cases (17%) in the general population (Koumans et al., 2007; Li et al., 2014; Jespers et al., 2014), (2) an association between previous BV infections and BV positive diagnostic (Bilardi et al., 2013; Guedou et al., 2013), (3) a higher risk factor for women using the pill instead of a condom (Bradshaw et al., 2013; Guedou et al., 2013), and (4) the history of previous pregnancy being higher in women with BV (Africa, Nel & Stemmet, 2014; Mengistie et al., 2014).

Table 1 Characteristics of the population of study (n = 150).

The samples classification as normal or BV was performed according the Nugent score.

Variables	Women with normal
flora (n = 124)	Women with BV
(n = 26)	
Age (years)	30.2 ± 11.42	32.5 ± 9.7	
With children (%)			
No	68.5	50.0	
Yes	27.4	50.0	
Pregnant women (%)	4.0	0.0	
Previously diagnosed with bacterial vaginosis (%)	16.9	38.5	
Contraception (%)			
No contraception	8.9	15.4	
Pill	54.0	61.5	
Condom	25.8	11.5	
Other	12.1	15.4	
Notes.

Data are mean ± standard deviation or n (%).

As shown in Table 2, the PNA-FISH method was able to diagnose 121 from a total of 124 healthy cases and was capable of categorizing 22 positive cases from a total of 26 BV cases when compared with the standard Nugent score. The PNA-FISH methodology was capable of illustrating clear differences between healthy and BV samples, showing specific detection of Lactobacillus spp. and G. vaginalis species directly in clinical samples. In fact, a typically healthy sample and a BV sample exhibited a totally different vaginal microflora, such as UM300 and UM235 samples, respectively, being clue cells, and G. vaginalis augmentation was easily detected in the UM235 sample (see Fig. 1). However, some discrepancies were also found between the two methodologies; specifically, in 7 vaginal samples. In fact, 4 vaginal samples were positive for BV by Gram staining but negative by PNA-FISH evaluation, while the others 3 vaginal samples were negative for BV by Gram staining but positive by PNA-FISH evaluation. It is well known that conventional BV diagnosis accuracy is highly dependent on the training and experience of the technician due to the unspecific staining of the Gram method (Simoes et al., 2006), which might explain some of the discrepant results observed.

Figure 1 Fluorescence microscopy pictures of Lactobacillus spp., Gardnerella vaginalis and others bacteria species from a healthy (UM300) and a BV (UM235) vaginal clinical samples by specific PNA probes (Lac663 and Gard162) associated with Alexa Fluor 488 and 594 fluorochromes and DAPI staining, respectively.

(A) Green filter; (B) red filter; (C) blue filter; (D) overlay of the three previous filters. As shown in the green filter (A), UM300 (healthy) and UM235 (BV) samples showed the presence of Lactobacillus spp. species but only BV sample demonstrated an elevated G. vaginalis concentration in the vaginal swabs (red filter (B)), which they proved to stablish clue cells by overground the vaginal epithelial cells in the blue filter (C). Therefore, both vaginal swab samples exhibited a totally different vaginal microflora, as finally we may observe in the overlay of the filters (D), being clue cells, and G. vaginalis augmentation was easily detected in the UM235 sample.

Table 2 Comparison between PNA-FISH method versus Gram staining using Nugent score criteria for BV diagnosis.

PNA-FISH results	Nugent results	
	BV+	BV+	Total	
BV+	22	3	25	
BV−	4	121	125	
Total	26	124	150	
Statistical analysis of PNA-FISH method	
	Estimated value	Lower limit	Upper limit	
Sensitivity	84.6%	64.3%	95.0%	
Specificity	97.6%	92.6%	99.4%	
Accuracy	95.3%	89.2%	98.3%	
Positive likelihood	34.97	11.30	108.24	
Negative likelihood	0.16	0.06	0.39	

To better evaluate the diagnostic value of the proposed PNA-FISH approach, the technique performance was assessed by determining the following parameters: specificity, sensitivity, accuracy, PLR, NLR. Based on these results, an experimental specificity of 97.6% (95% CI [92.6–99.4%]) and sensitivity of 84.6% (95% CI [64.3–95.0%]) were obtained for the BV diagnosis by our PNA-FISH method (Table 2). Therefore, a high accuracy was also obtained for our PNA-FISH method; more exactly, a value of 95.3% (95% CI [89.2–98.3%]).

Regarding the likelihood ratios, the PNA-FISH method evidenced a PLR of 34.97 and a NLR of 0.16. So, the specificity and the NLR values show the test ability to correctly identify as normal a person who does not have BV. Meanwhile, the low NLR obtained, in fact, shows that the probability of having BV is much decreased (0.16) for a negative PNA-FISH result. Moreover, our experimental specificity is revealed to be superior to Nugent’s Gram stain system specificity (83%) (Schwebke et al., 1996). Therefore, our method was able to correctly identify 97.6% of those patients previously classified with normal vaginal flora, making PNA-FISH a trustful method to ensure a healthy diagnosis and avoiding false positive results.

In opposition, the sensitivity and PLR values demonstrated a strong association between a positive result for BV diagnostic and the probability of the patient indeed having BV. In this case, the high PLR shows us the increase in probability of having BV (35×) if the test result is positive. The sensitivity value was in fact lower than expected, taking in consideration our previous in vitro experiments, where we have reached to a sensitivity of 100% (95% CI, [81.5–100.0%]) (Machado et al., 2013). Despite the fact that the experimental sensitivity (84.6%) was slight lower than the specificity of the Gram staining by the Nugent score (89%) (Schwebke et al., 1996), it was nevertheless higher than the Amsel criteria sensitivity (60%) determined by Gallo et al. (2011). It is important to mention that other bacterial species, with similar Gram staining morphology, could be at a high number in the samples, leading to an incorrect classification of BV according to Nugent criteria. In fact, Verhelst and colleagues presented evidences that infers a lack of accuracy in the interpretation of the results in Gram stain by the Nugent score in their clinical results (Verhelst et al., 2005). Forsum and colleagues also found discrepancies in scoring bacterial cell types when pleomorphic lactobacilli and other kinds of bacteria could be regarded as G. vaginalis cells, leading to an incorrect BV diagnosis (Forsum et al., 2002; Schwiertz et al., 2006). Also, it is important to mention that G. vaginalis may vary in size and form, from round to more elongated, with no defined border to separate them from the lactobacilli morphotypes (Forsum et al., 2002), thus illustrating again problems in the accuracy of the smears interpretation. These facts suggest that the sensitivity value has likely been underestimated.

Overall, despite the cost-effective nature of the Nugent score, the PNA-FISH appears to be an accurate method for detecting BV from vaginal samples while maintaining a similar complexity as the previous standard method.

Conclusions

In conclusion, in this study we described the first PNA-FISH methodology applied for BV diagnosis, and the parameters evaluated have proved its potential as a diagnostic tool. The performance characteristics of this PNA-FISH method also suggest that it might be a reliable alternative to the Amsel criteria and Gram stain under the Nugent score. Despite that our sample size was somewhat small, the population at study was representative from what has been described by many other epidemiological studies, therefore validating this prospective study.

Supplemental Information

Supplemental Information 1 Raw data associated with this manuscript

Click here for additional data file.

Additional Information and Declarations

Competing Interests

Author Contributions

Human Ethics

António Machado, Carina Almeida and Nuno Cerca have a patent on the PNA-FISH probe described in this paper (Portuguese patent office (INPI20121000090117)).

António Machado performed the experiments, analyzed the data, contributed reagents/materials/analysis tools, wrote the paper, prepared figures and/or tables, reviewed drafts of the paper.

Joana Castro performed the experiments, reviewed drafts of the paper.

Tatiana Cereija performed the experiments, analyzed the data, reviewed drafts of the paper.

Carina Almeida analyzed the data, contributed reagents/materials/analysis tools, reviewed drafts of the paper.

Nuno Cerca conceived and designed the experiments, wrote the paper, reviewed drafts of the paper.

The following information was supplied relating to ethical approvals (i.e., approving body and any reference numbers):

Subcomissão de Ética para as Ciências da Vida e Saúde, from Universidade do Minho (process SECVS 003/2013).

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
