# Peer review of "Diagnosis of bacterial vaginosis by a new multiplex peptide nucleic acid fluorescence in situ hybridization method"

_PeerJ, doi:10.7717/peerj.780_

## Round 0.1 · original submission · Minor Revisions

As you will see, both expert reviewers consider your manuscript a well written technical study that adds valuable approaches to the diagnosis of BV. Both reviewers provide some suggestions for further improvement of the paper. In addition, I would suggest you use the term microbiota rather than flora/microflora.

Reviewer 1 ·

Basic reporting

The paper is a technical paper. With some minor translational issues, it is well written and concise. The only technical comment I have is the use of the transport media to generate a Nugent slide. Some readers will state that the media may have a dilution effect on the specimen placed on the slide and could potentially alter a Nugent score. They will need a reference to support the approach, or a sentence discussing this as an experimental limitation.

My comments--

Abstract:
1st sentence: “infection” to “infections”.
1st sentence: run-on sentence, split into two sentences.
2nd sentence: “gold standard” to “research gold standard”.
3rd sentence: “characteristic” to “characteristics”.

Paper:
Line 49: “being” to “with”.
Line 49: “Fluorescence” to “fluorescence”.
Line 52: remove “Nowadays,”.
Lines 174 and 175: “sensibility” to “sensitivity”.
Table 1 is demographically interesting, but not really necessary to the content of the paper.
Table 1: add “With” to “Children”.

Delete Table 1 - it is confusing and adds nothing to the paper

Experimental design

no issues

Validity of the findings

no issues

Additional comments

see above

·

Basic reporting

The article is an original research that deserves to be published. Provides clinical data provide a novel diagnostic method. Figure recommend that have a lighter walk figure to explain each of the panels containing.

Experimental design

The experimental design seems appropriate

Validity of the findings

The data are robust, it worths to be published

Additional comments

This is a original manuscript that should be published, minor review in figure caption must be done and minor review in the references

---

## Round 0.2 · accepted · Accept

You adequately addressed all suggestions made by the reviewers of the original manuscript.